# Enhancing the Antibody Production Efficiency of Chinese Hamster Ovary Cells through Improvement of Disulfide Bond Folding Ability and Apoptosis Resistance

**DOI:** 10.3390/cells13171481

**Published:** 2024-09-04

**Authors:** Chen Zhang, Yunhui Fu, Wenyun Zheng, Feng Chang, Yue Shen, Jinping Niu, Yangmin Wang, Xingyuan Ma

**Affiliations:** 1State Key Laboratory of Bioreactor Engineering, School of Biotechnology, East China University of Science and Technology, Shanghai 200237, China; zhang1990chen@126.com (C.Z.);; 2Shanghai Key Laboratory of New Drug Design, School of Pharmacy, East China University of Science and Technology, Shanghai 200237, China

**Keywords:** CHO cells, antibody production efficiency, disulfide bond folding, apoptotic resistance, unfolded protein response

## Abstract

The complex structure of monoclonal antibodies (mAbs) expressed in Chinese hamster ovary (CHO) cells may result in the accumulation of unfolded proteins, triggering endoplasmic reticulum (ER) stress and an unfolded protein response (UPR). If the protein folding ability cannot maintain ER homeostasis, the cell will shut down protein translation and ultimately induce apoptosis. We co-overexpressed HsQSOX1b and survivin proteins in the antibody-producing cell line CHO-PAb to obtain a new cell line, CHO-PAb-QS. Compared with CHO-PAb cells, the survival time of CHO-PAb-QS cells in batch culture was extended by 2 days, and the antibody accumulation and productivity were increased by 52% and 45%, respectively. The proportion of (HC-LC)_2_ was approximately doubled in the CHO-PAb-QS cells, which adapted to the accelerated disulfide bond folding capacity by upregulating the UPR’s strength and increasing the ER content. The results of the apoptosis assays indicated that the CHO-PAb-QS cell line exhibited more excellent resistance to apoptosis induced by ER stress. Finally, CHO-PAb-QS cells exhibited mild oxidative stress but did not significantly alter the redox status. This study demonstrated that strategies based on HsQSOX1b and survivin co-overexpression could facilitate protein disulfide bond folding and anti-apoptosis ability, enhancing antibody production efficiency in CHO cell lines.

## 1. Introduction

Chinese hamster ovary (CHO) cells have been the most prevalent production cell line for the past 30 years and continue to be the industry’s primary workhorse for therapeutic protein production [1]. To meet the growing global demand for biopharmaceuticals, especially therapeutic antibodies, CHO cells are needed to improve their production efficiency further and reduce production costs [2,3]. The synthesis of secreted proteins by CHO cells involves a series of processes, including transcription, translation, post-translational modification, protein folding, and secretion. The endoplasmic reticulum (ER) provides a distinctive oxidative milieu that is indispensable for the optimal operation of the enzymes and chaperones that are responsible for catalyzing the reactions that modify, fold, and assemble nascent proteins to obtain the functional conformation necessary for transport to the Golgi apparatus [4]. It has been demonstrated in multiple studies that product yield is not invariably proportional to gene copy number and mRNA levels [5]. The post-translational process is the bottleneck for secreted protein production [6]. Oxidative protein folding, characterized by intramolecular disulfide bond formation, is arguably the most complex protein folding problem and a rate-limiting step in synthesizing secreted proteins [7]. The ER acts as the biosynthetic organelle of all secreted proteins and membrane proteins, and excessive and complex protein production can lead to protein unfolding and misfolding, which triggers ER stress [8]. At the same time, the cells trigger the unfolded protein response (UPR) to both maintain and restore homeostasis in the ER by weakening protein translation, activating folding mechanisms, and degrading the unfolded protein [9]. However, if ER homeostasis cannot be maintained due to limited protein folding capacity, the cell shuts down protein translation to minimize the ER load and ultimately induce apoptosis [10,11]. Therefore, it is crucial to fine-tune and control the UPR to improve the performance of the ER and maintain cell homeostasis to enhance the productivity and robustness of CHO cells [12]. These are also the current challenges in CHO cell engineering.

Monoclonal antibodies (mAbs) possess a complex structure and function compared to other recombinant proteins produced in CHO cells [13]. Most therapeutic mAbs belong to IgG1, a subclass of immunoglobulin G (IgG), which is a heterodimer molecule of about 150 kDa consisting of two heavy chains (HC, 50 kDa) and two light chains (LC, 25 kDa) [14]. For disulfide bond folding, HCs and LCs are translocated to the ER; HCs and LCs co-translate the translocations to the ER for disulfide bond folding [15]. A complete IgG1 molecule contains 16 disulfide bonds, and the correct folding of disulfide bonds enables IgG1 molecules to form the correct conformation, maintain the integrity of IgG1 molecules, and maintain their biological activity [16]. Therefore, the folding ability of disulfide bonds is essential in antibody-producing cells. In the ER of CHO cells, the oxidation capacity of molecular oxygen is utilized mainly by protein disulfide isomerase (PDI) and endoplasmic reticulum oxidoreductase 1 (Ero1) to generate new disulfide bonds in the newly folded protein [4,17]. PDI has previously been actively studied to study the folding efficiency of disulfide bonds within CHO cells. For instance, when PDIa4 was knocked down in CHO cells, it was observed that the amount of antibody secretion was reduced, and the accumulation of immature antibodies in the cells was observed [18]. At the same time, the addition of recombinant PDIa4 could refold the antibodies and Fas. Nevertheless, the overexpression of PDI has yielded disparate outcomes in the production of recombinant protein products, which is contingent upon the cell line, model protein products, and cell engineering target genes. A study has demonstrated that the overexpression of PDI resulted in a notable enhancement in the secretion rate of human antibodies, with an increase of approximately 27% [19]. Conversely, intracellular retention of TNFR/Fc protein was observed in other recombinant CHO cells, and PDI overexpression did not impact the productivity of thrombopoietin-secreting CHO cells [20,21]. Since PDI is dependent on Ero1 for sustained disulfide bond folding, and the rate-limiting step of disulfide bond formation is the oxidative regeneration of PDI in the reduced state, the researchers have considered Ero1 expression level to be an essential factor in antibody disulfide bond formation and have investigated this [22]. Mohan et al. showed the transient overexpression of Ero1 and co-expression of Ero1 and PDI in antibody-producing rCHO cells, which increased the specific antibody growth rate by 37% and 55%, respectively [23]. In the process of stable overexpression of Ero1 and PDI, it was found that antibody expression did not achieve the expected increase effect, which suggested that the productivity may depend on the expression level of PDI and the ratio of Ero1 to PDI.

ER-mediated apoptosis is triggered by sustained ER stress via UPR, and apoptotic cells rapidly enter the death stage, producing contaminated cell debris, shortening culture time, and reducing production and product quality [24]. Cell engineering methods to inhibit or delay apoptosis have been developed in CHO cells, focusing on manipulating pro-apoptotic and anti-apoptotic proteins [25]. Heat shock proteins (HSPs) are produced in response to low-stress levels in what is known as a “stress response,” with many family members acting as companions for the unfolded protein and generally acting as stabilizers for the folded protein [26]. For example, HSP27 can inhibit the caspase-dependent arms of Fas-activated exogenous pathways and caspase-3 activation. Another member of the HSP family [27], HSP70, inhibits mitochondria-related apoptosis by inhibiting BAX transport from the cytosol to the mitochondria [28]. There are also many studies on the anti-apoptotic effect of BCL-2 family members through CHO cell engineering. The general research idea is the overexpression of negative apoptotic regulators (BCL-2, BCL-xL, and MCL-1) and the knockout of positive apoptotic regulators (BAK and BAX) [29,30,31,32]. Most results have demonstrated extended cell culture times and titer increases in products ranging from 35% to 500% [31,33]. As caspase is a pivotal factor in signaling during caspase-dependent apoptosis, it is a logical target for anti-apoptotic engineering [25]. Current studies have targeted caspase-3, -7, -8, and -9 in CHO cells through knockdown action or overexpression of caspase inhibitors XIAP and CRMA [24,34,35]. Caspase knockdown studies in CHO cells typically result in a slight increase in viable cell density or survival, of up to 40% [36]. These anti-apoptotic cell engineering strategies help combat the apoptosis caused by ER stress, maintain cell homeostasis, and positively affect the productivity of recombinant products.

Quiescin sulfhydryl oxidase 1 (QSOX1) is a unique thiol oxidase that can form disulfide bonds and transfer disulfide simultaneously. It is widely distributed in all multicellular organisms and can introduce disulfide bonds to substrate proteins independently of Ero1 [37,38]. When the tri (2-carboxyethyl) phosphine and dithiothreitol were used as substrates, the enzyme activity of HsQSOX1b was 100 times higher than that of Ero1 and Erv family sulfhydryl oxidase [39]. Survivin is the smallest member of the inhibitor of the apoptosis protein (IAP) family and possesses the baculovirus apoptosis inhibitor protein repeat (BIR) structure [40]. Survivin has been demonstrated as inhibiting caspase-3/7 activity, thereby inhibiting apoptosis directly [41]. Additionally, it has been shown to inhibit apoptosis induced by Fas, Bax, and anticancer drugs, indicating a broad range of anti-apoptotic capabilities [42]. Previous studies conducted by our research group have demonstrated that the stable overexpression of the HsQSOX1b and survivin genes in CHO-K1 cells can enhance the folding ability of the disulfide bond of humanized Gaussia luciferase (GLuc), improve anti-apoptosis ability, and prolong the protein production cycle of cells [43]. In this study, we will further investigate the effect of overexpression of HsQSOX1b and survivin on antibody efficacy in antibody production strains and investigate the mechanism in the cell. Our study suggests that the cellular strategy of HsQSOX1b and survivin co-overexpression can improve the ER folding ability and anti-apoptosis ability of antibody-producing cell lines and the production of complex recombinant proteins in CHO cell lines.

## 2. Materials and Methods

### 2.1. Plasmid Construction

The preliminary laboratory constructed the expression box EC#2, which contains the coding genes for HsQSOX1b and survivin. HsQSOX1b is the abbreviation of human quiescin sulfhydryl oxidase 1 isoform b, in which the secretion of the precursor peptide was removed. The N-terminal end of HsQSOX1b was connected to the fluorescent protein EGFP, and an ER localization sequence, KDEL, was added to the C-terminal end, referred to as EQK for short. It is linked to survivin by a self-clipped peptide, T2A. PCR amplified the EC#2 expression box with the restriction sites of *Hind* III and *BamH* I added and was recombined into the mammalian expression vector pcDNA3.1(+) plasmid, as shown in Appendix A, and briefly named pcDNA3.1(+)-QS.

### 2.2. Cell Line Development

In this study, we engineered the host of CHO-PAb, a cell line that stably expresses pembrolizumab antibodies. The pembrolizumab antibody gene sequence in CHO-PAb cells is stably integrated into the active transcription site *H11* and has been shown to have stable expression properties of pembrolizumab antibodies. Further information on CHO-PAb cell lines is available in a previous publication [44]. The CHO-PAb cells were digested and spread on 6-well plates one day in advance, with approximately 5 × 10^5^ cells added to each well, and fresh medium was replaced the next day. The transfer solution was prepared by combining 100 μL of opti-MEM medium (Gibco Life Technologies, Gaithersburg, MD, USA), 2 μg of pcDNA3.1(+)-QS plasmid, and 4 μL of Lipo8000 transfection reagent (Beyotime, Shanghai, China), and gently mixing the components. The transfer solution was then added dropwise to the 6-well plate, gently rolled to mix well, and cultured in a CO_2_ incubator.

Following 48 h of transfection, stably integrated cell lines were identified using a medium containing 2 mg/mL of G418 antibiotic, with the medium being replaced every 2–3 days. Monoclonal cells were then obtained from cell pools through a limited dilution process. The cells digested with trypsin were diluted in a medium and inoculated at a density of 0.5 cells per well in 96-well plates. After incubation for approximately two weeks, cells with robust growth and green fluorescence excitation were selected by fluorescence inverted microscope (Olympus, Nagano, Japan) and transferred to 6-well plates for further culture and analysis.

Cell proliferation experiments were performed using the xCELLigence RTCA instrument (Agilent, Santa Clara, CA, USA). A total of 2000 target cells were enumerated via plate counting, diluted with the same medium of 200 μL, and added to E-Plate 16. The cells were then operated according to the manufacturer’s instructions of the RTCA instrument, which automatically records the cell growth trend.

### 2.3. Cell Culture

Adherent cell culture: The CHO series of adhesion cells were cultured using Ham’s F-12 medium (Gibco Life Technologies) with 10% fetal bovine serum and 1% penicillin-streptomycin amphotericin (Solarbio, Beijing, China) in a cell incubator at 37 °C in a humid environment of 5% CO_2_. Cells with a fusion of 80–90% were digested with 0.25% trypsin (Solarbio) and cultured in a ratio of 1:4, with the medium changed every two days.

Suspension cell culture: The CHO series cells were acclimated in suspension by gradually decreasing serum. This involved reducing the serum in Ham’s F-12 medium from 10% to 0.5%, mixing it with CHOGrow^®^ CD serum-free medium (Basalmedia, Shanghai, China) five times, and then transferring the cells to suspension culture bottles. The culture was continued with a serum-free medium at 37 °C, 5% CO_2_, and 120 rpm. Cell count was performed with a cell counting apparatus (Thermo Fisher Scientific, Waltham, MA, USA), and the cell density reached 2 × 10^6^ cells/mL or more, which was in the logarithmic growth stage. Cell passage could be performed according to the living cell density of 3 × 10^5^ cells/mL.

### 2.4. Cell Parameter Detection during Batch Culture

Detection of living cell density and cell viability: A total of 25 mL CHO-PAb and CHO-PAb-QS suspension cells with a density of 5 × 10^5^ cells/mL were inoculated into a 125 mL suspension culture bottle which was maintained at 37 °C, 5% CO_2_, and 120 rpm. Three parallel groups were set up daily to detect the live cell density and cell viability by cell counter.

ELISA to detect the antibody content: A total of 0.2 mL of the cell suspension was removed and centrifuged at 12,000 rpm for 5 min, and the culture supernatant was collected daily. ELISA plates were coated with 100 μL of 2 mg/mL PD-1 antigen (DIMA BIOTECH, Wuhan, China) per well and incubated overnight at 4 °C. Subsequently, non-specific binding was blocked with 1% BSA for 1 h. Samples of the pembrolizumab standard (DIMA BIOTECH) and diluted culture supernatant were then added to the appropriate pore for 1 h with a PBST wash plate. Following the addition of HRP-Goat Anti-Human IgG (H + L) (Proteintech, Wuhan, China) and incubation for 1 h at room temperature, the TMB substrate solution (Solarbio) was added. The absorbance at 405 nm was then detected by the microplate reader (Thermo Fisher Scientific).

Calculation of cell line unit productivity: The suspension cell antibody production rate (q_mAb_, pg/cell/day) was calculated by the following formula:$$q_{mAb}=\frac{m_{mAb}}{ICA}$$

m_mAb_ is the total mass of mAb in the culture supernatant measured by ELISA and ICA (integral cell area) is the overall cell area. The calculation formula is as follows:$$ICA=\frac{\left(N-N_{0}\right)\times t}{\log_{e}\left(N/N_{0}\right)}$$
where N and N_0_ are the final and initial number of living cells and t is the number of days of culture.

### 2.5. Western Blot

Methods of obtaining exocrine antibodies and cell lysis are described in detail in our previous publications [44]. The folding state of the antibody was analyzed by denaturing non-reducing SDS-PAGE electrophoresis and Western blot. The T25 culture flask was inoculated with 2 mL of 5 × 10^5^ cell suspensions. After 24 h of adherence culture, the culture medium was replaced with fresh medium or medium containing 20 μg/mL BFA (Beyotime), and the total protein was collected after 8 h of further incubation for Western blot analysis. The SDS-PAGE employed was 4–20% Bis-Tris precast page gel (Adamas, Shanghai, China) with a voltage of 180 V, running to the bottom of the page for approximately 50 min. Denatured reduction SDS-PAGE electrophoresis was employed to examine ER stress marker proteins, internal reference proteins, and antibodies in extracellular media. The SDS-PAGE utilized was 12% Tris-Glycine precast page gel (Adamas). For the Western blot analysis of antibodies, the primary antibody was not incubated, and the secondary antibody, HRP-Goat Anti-Human IgG (H + L) (Proteintech), was incubated for 1 h after closure and cleaning with a dilution ratio of 1:10,000.

For the Western blot analysis of ER stress marker proteins, the primary antibody was used in conjunction with the following antibodies: PERK (125 kDa, 1:2000, Abmart, Shanghai, China); BiP (78 kDa, 1:2000, Abmart); CHOP (27 kDa, 1:2000, Abmart); and β-actin (43 kDa, dilution ratio 1:8000, Abcam, Cambridge, MA, USA). The secondary antibodies employed were HRP-Goat Anti-Rabbit IgG (H + L) (Proteintech) and HRP-Goat Anti-Mouse IgG (H + L) (Proteintech), both with a dilution ratio of 1:10,000.

### 2.6. ER and Golgi Content Detection

Then, 2 × 10^6^ cells were collected and washed with the PBS solution. The cells were centrifuged at 800 rpm for 5 min to remove the washing solution, leaving the cell precipitation. The cells were then dispersed with 1 mL of ER-Tracker Red or Golgi-Tracker Red (Beyotime) diluent and incubated for 30 min. The ER-Tracker Red or Golgi-Tracker Red staining solution was removed by centrifugation at 300× *g* for 5 min, and the cells were washed with PBS solution twice. After that, the cells were analyzed by flow cytometry (Beckman Coulter, Brea, CA, USA).

### 2.7. Detection of Cell Apoptosis Rate

Annexin V-PE/7-AAD apoptosis detection kit (Vazyme, Nanjing, China) was used for the determination. Cells were cultured with 10 μg/mL puromycin or 2.5 μg/mL tunicamycin for 48 h. The cells were harvested and supplemented with 5 μL Annexin V-PE and 5 μL 7-AAD staining solution. The cells were gently washed and incubated for 10 min at room temperature without light. A total of 200 μL of 1× binding buffer was added, gently and evenly aspirated, and flow cytometry was performed. Tens of thousands of cells were collected for each sample. The excitation wavelength for flow cytometry was 488 nm. The fluorescence of PE was detected in the FL2 channel, and the fluorescence of 7-AAD was detected in the FL3 channel. The apoptosis rate was analyzed using FlowJo software (FlowJoV10, Becton, Dickinson & Company, New York, NY, USA).

### 2.8. Detection of Intracellular Caspase-3 Activity

The caspase-3 activity assay kit (Beyotime) was used. Cells were cultured with 10 μg/mL puromycin or 2.5 μg/mL tunicamycin for 48 h, cell precipitates were collected, cells were suspended with 150 μL lysate, cells were lysed in an ice bath for 15 min, centrifuged at 4 °C at 16,000× *g* for 15 min, and the lysate supernatant was collected in a new precooled centrifuge tube. The protein concentration in the cleavage supernatant was determined using the Bradford method. A total of 100 μL ρNA standard of 0, 10, 20, 50, 100, and 200 μM was added to the enzyme-labeled plate, and the enzyme-labeled instrument measured the absorbance of 405 nm, and the standard ρNA curve was plotted. Then, 50 μL of the cleavage supernatant to be measured was added to 40 μL of detection buffer, and 10 μL of Ac-DEVD-ρNA at a concentration of 2 mM was added. After incubation at 37 °C for 2 h, the absorbance of 405 nm was measured on the microplate reader. The amount of ρNA catalyzed by the sample was obtained according to the calculation formula of the standard curve. According to the number of caspase-3 enzyme activity units and the protein concentration of the cleavage supernatant, the enzyme activity units of caspase-3 contained in the protein per unit weight of each sample could be calculated as μmol/mg or mmol/g.

### 2.9. Detection of Total Antioxidant Capacity of Cells

A ferric reducing ability of plasma (FRAP) kit (Beyotime) was used to detect the total antioxidant capacity of cells. A total of 1 × 10^6^ cells were collected, frozen, and thawed repeatedly in liquid nitrogen and a 37 °C water bath, centrifuged at 12,000× *g* at 4 °C for 10 min; the supernatant was collected, and the total protein concentration was determined by the BCA protein concentration detection method. Then, 100 mM FeSO_4_ solution was diluted with distilled water to 0.15, 0.3, 0.6, 0.9, 1.2, and 1.5 mM as standard solutions. Each well of the 96-well plate was filled with 180 μL FRAP working fluid, 5 μL FeSO_4_ standard solution of various concentrations was added to the standard curve test well, and 5 μL distilled water was added to the blank control well. Samples of cell lysis supernatant were added to the sample test well. After incubation at 37 °C for 5 min, the 593 nm absorbance value of each well was determined by a microplate reader, and the total antioxidant capacity of the sample was calculated according to the standard curve. The total antioxidant capacity of the sample is expressed as the concentration of the FeSO_4_ standard solution. Finally, the total antioxidant capacity per milligram or gram of protein was calculated in mmol/mg or mmol/g.

### 2.10. Measurement of GSH and GSSG

The GSH and GSSG assay kit (Beyotime) measured the GSH and GSSG in cells. GSSG can be reduced to GSH by glutathione reductase, and GSH can react with the decolorizing substrate DTNB to produce yellow TNB and GSSG. When these two reactions are combined, tGSH (GSSG + GSH) acts as a rate-limiting factor for color production, and the amount of tGSH determines the amount of yellow TNB formed. Fresh cells were harvested, protein removal reagent M solution was added at 3× cell precipitation volume, frozen and thawed repeatedly in liquid nitrogen and 37 °C water bath, centrifuged at 12,000× *g* at 4 °C for 10 min, supernatant was collected, and total protein concentration was determined by BCA protein concentration detection method. To each 100 μL GSSG standard solution and supernatant sample, 20 μL GSH scavenging auxiliary solution and 4.8 μL scavenging reagent working solution were added, immediately mixed, and reacted at 25 °C for 60 min. To the 96-well plate, 10 μL standard or sample was added sequentially, combined with 150 μL total glutathione detection working solution, and incubated at 25 °C or room temperature for 5 min. After mixing with NADPH solution for 25 min, the microplate reader used the absorbance of 412 nm. Calculation of the tGSH content in the sample: the sample can be calculated against the standard curve to tGSH (GSSG concentration calculated by the standard curve multiplied by 2) or GSSG content; with the amount of tGSH minus the content of GSSG, you can calculate the content of GSH.

### 2.11. Transcriptome Analysis

RNA sequencing was performed on the CHO-PAb-QS cell line using the CHO-PAb cell line as a reference. Total RNA was extracted using the TRIzol reagent (Invitrogen, Carlsbad, CA, USA) according to the manufacturer’s protocol. Then, the libraries were constructed using VAHTS Universal V6 RNA-seq Library Prep Kit according to the manufacturer’s instructions. The transcriptome sequencing and analysis were conducted by OE Biotech Co., Ltd. (Shanghai, China). The DESeq2 software (version 1.34.1) was employed to perform differential expression gene analysis, with a focus on the protein processing in the ER (KEGG pathway map04141), the apoptosis pathway in mammals (KEGG pathway map04215), and the ER-Golgi vesicular transport pathway (KEGG pathway map04130). The results were presented in a color-coded format, with the log_2_Foldchange value as the basis for the color scheme (*p*-value < 0.05).

### 2.12. Statistical Analysis

All experimental data were presented as mean ± SD and statistical analysis and graphing were performed using GraphPad Prism 9 software. Data from each group were compared by *t*-test or one-way analysis of variance test; *p* < 0.05 was a significant difference.

## 3. Results

### 3.1. Construction of CHO-PAb-QS Cell Line with HsQSOX1b and Survivin Co-Overexpression

We stably integrated pcDNA3.1(+)-QS plasmid into the genome of stable producer CHO-PAb cells secreting pembrolizumab antibodies to obtain a CHO-PAb-QS cell pool that overexpressed both EGFP-HsQSOX1b-KDEL (EQK) and survivin proteins (Figure 1A). After intracellular expression, EQK and survivin are dissociated, with EQK localized in the ER and survivin dispersed in the cytoplasm (Figure 1A). We selected two robust CHO-PAb-QS-a5 and CHO-PAb-QS-b3 monoclonal cell lines and observed them under the fluorescence microscope. The HsQSOX1b protein, fused with the EGFP protein and the ER localization sequence KDEL, exhibited green fluorescence and a dark nucleus (Figure 1B). The expression of QSOX and survival was validated by Western blot analysis (Figure 1C), which demonstrated the background expression of QSOX1 (67 kDa) and the overexpression of EQK protein (96.5 kDa), as well as the overexpression of survivin (16 kDa). The RTCA instrument detected the proliferation curves of two monoclonal cell lines, shown in Figure 1D. The growth of the CHO-K1 cell line was significantly faster than the other three cell lines. The CHO-PAb-QS-a5 and CHO-PAb-QS-b3 cell lines loaded more functional genes than CHO-PAb, resulting in a slightly slower proliferation rate. However, the growth trends of these three antibody-producing cell lines and the time to reach the stable stage were similar. Given that the CHO-PAb-QS-b3 cell line exhibited a more significant overexpression of proteins than the CHO-PAb-QS-a5 cell line and demonstrated a more stable growth pattern, the CHO-PAb-QS-b3 cell line was selected for subsequent experiments with CHO-PAb, which was designated as CHO-PAb-QS.

### 3.2. Cell Growth and Antibody Expression during Batch Culture

The suspension domestication of CHO-PAb and CHO-PAb-QS cells was performed to simulate industrial production. CHO-PAb and CHO-PAb-QS suspension cells were cultured in batches for about two weeks, and viable cell density (VCD), cell viability, and accumulated antibody production were measured daily. During the initial five days of the culture period, the cell viability of CHO-PAb and CHO-PAb-QS decreased slowly but remained above 90% (Figure 2A). As nutrients were consumed and harmful substances accumulated, cell viability declined significantly from the sixth day onwards. By the tenth day, the viability of CHO-PAb cells had fallen to 5.8%. In contrast, the decline in CHO-PAb-QS cell viability was slower. The cell viability rate was 31.9% on the 10th day, and the experiment was terminated when the cell viability dropped below 10% on the 12th day. On day 6 of the batch culture, both cell lines reached the maximum viable cell density (VCD_max_). The VCD_max_ of CHO-PAb-QS cells was 3.4 × 10^6^ cell/mL (Figure 2B), 8 × 10^5^ cell/mL lower than CHO-PAb cells (4.2 × 10^6^ cell/mL). The VCD of the two cell lines decreased significantly after the 6th day of culture, the decline in CHO-PAb cells was more rapid, and the VCD was as low as 2 × 10^4^ cell/mL on the 10th day, resulting in no further detection. However, the VCD of CHO-PAb-QS cells was 9.1 × 10^5^ cells/mL on day ten and 1.55 × 10^5^ cells/mL on 12th day. Compared with CHO-PAb cells, co-overexpression of EQK and survivin decreased the VCD_max_ of CHO-PAb-QS cells by about 20%, but the cell viability period was prolonged by two days, indicating a longer stable period and longevity. This suggests that CHO-PAb-QS cells can maintain a stable culture for a more extended period, have more robust anti-apoptotic properties, and have a longer lifespan when cultured in batches.

ELISA quantified the accumulation of antibodies by CHO-PAb and CHO-PAb-QS cells, and the antibody production rate of these two cell lines was calculated based on the VCD_max_ and antibody accumulation. From the 5th day of culture, the antibody accumulation of CHO-PAb-QS cells exceeded that of CHO-PAb cells, and the accumulated antibody concentration reached 241 μg/mL after 12 days of culture. However, CHO-PAb cells were cultured only until the 10th day due to low live cell density and cell viability, and the antibody accumulation was 159 μg/mL (Figure 2B). The cell production rate was calculated according to the time of VCD_max_, that is, the accumulation of antibodies on the 6th day (Figure 2C). CHO-PAb cells were 4.7 pg/cell/day, while CHO-PAb-QS cells reached 6.8 pg/cell/day, about 1.45 times that of CHO-PAb cells. These results showed that CHO-PAb-QS cells had higher antibody accumulation, antibody production rate, and production efficiency than CHO-PAb cell lines in batch culture.

### 3.3. Folding Analysis of Intracellular Antibodies

SDS-PAGE and Western blot experiments were conducted on antibodies derived from CHO-PAb and CHO-PAb-QS cells under non-reductive denaturation conditions. In this experiment, Brefeldin A (BFA) impeded the transport of antibodies to the Golgi apparatus in CHO-PAb and CHO-PAb-QS cells, resulting in the antibodies remaining in the ER. Western blot analysis revealed that CHO-PAb-QS cells exhibited higher levels of fully folded antibodies (HC-LC)_2_ compared to CHO-PAb cells (Figure 3A), accompanied by a reduction in the number of types and levels of incomplete folded antibodies (including HC_2_-LC, HC_2_, HC-LC, HC, and LC). The intervention of BFA resulted in the retention of antibodies produced by CHO-PAb and CHO-PAb-QS cells in the ER, with a concomitant reduction in the antibody content detected in the extracellular medium. The accumulation of antibody (HC-LC)_2_ in CHO-PAb-QS cells was more pronounced than in CHO-PAb cells. The gray values of antibody bands in the graph were calculated using Image J software (V1.8.0.112, National Institutes of Health, Bethesda, USA) to evaluate the proportion of fully folded antibodies (HC-LC)_2_ in all antibodies (Figure 3B). Under normal antibody secretion conditions, the proportion of CHO-PAb-QS intracellular (HC-LC)_2_ was 40.82%, nearly double that of CHO-PAb cells (22.35%). In the experiment with the addition of BFA, the proportion of (HC-LC)_2_ in CHO-PAb-QS cells increased to a staggering 73.47%, while CHO-PAb cells only rose from 22.35% to 49.63%. According to these results, it can be inferred that CHO-PAb-QS cells have a higher disulfide bond folding speed than CHO-PAb cells.

### 3.4. Expression Analysis of ER Stress Marker Proteins in Cells

ER stress markers PERK, BiP, and CHOP levels were also quantified. The results showed that the expressions of PERK, BiP, and CHOP proteins in CHO-PAb-QS cells were higher than those in CHO-PAb cells (Figure 4A). In the absence of any external stimuli, the relative expression of PERK in CHO-PAb-QS cells was 1.29 times that of CHO-PAb cells, the relative expression of BiP was 1.79 times that of CHO-PAb cells, and the relative expression of CHOP was 1.21 times that of CHO-PAb cells. In this data set, CHO-PAb-QS cells’ UPR intensity was higher than CHO-PAb cells. The addition of BFA resulted in a significant increase in the expression of PERK, BiP, and CHOP proteins in CHO-PAb and CHO-PAb-QS cells (Figure 4B). The expressions of PERK, BiP, and CHOP protein in CHO-PAb cells increased by 1.29 times, 1.56 times, and 1.32 times, respectively, while those in CHO-PAb-QS cells increased by 1.15 times, 1.06 times, and 1.43 times, respectively. Following BFA treatment, the PERK, BiP, and CHOP protein expression levels of CHO-PAb-QS cells were 1.13, 1.22, and 1.31 times that of CHO-PAb cells. These results indicated that CHO-PAb-QS cells exhibited a more robust ER load and were better able to maintain ER homeostasis than CHO-PAb cells. In conjunction with the analysis of antibody folding in CHO-PAb and CHO-PAb-QS cells, the comparison of relative expression levels of ER stress proteins revealed that in response to elevated folding speeds and protein productivity within the ER, CHO-PAb-QS cells augmented the load of the ER by upregulating the UPR intensity, thereby ensuring cellular vitality and productivity.

RNA-seq transcriptome analysis was performed on the CHO-PAb-QS cell line using the CHO-PAb cell line as a reference. The results of the KEGG enrichment analysis indicated that the KEGG pathway (map04141) exhibited a relatively complete array of ER activities, encompassing the correct protein processing, endoplasmic reticulum-associated degradation (ERAD), and the UPR signaling pathway (Appendix A). A notable increase in the expression of genes involved in glycosylation was observed in the correct protein processing, including the GlcI and ERManI genes. It is speculated that due to the fast disulfide bond folding rate of CHO-PAb-QS cells, the intracellular glycosylation reaction is correspondingly improved. Among the three signaling pathways of the UPR response, the expression of many critical factors in the PERK and IRE1 pathways was elevated despite most of the genes in the ATF6 pathway having a *p*-value ≥ 0.05 and being out of the labeling range. The results indicated that the degree of the UPR signal activation and ER pressure in CHO-PAb cells was higher than that of CHO-PAb cells. In addition to the fact that the co-overexpression of EQK and survivin in CHO-PAb-QS cells would impose a specific load on the ER, the enhanced folding capacity of disulfide bonds and the accelerated synthesis of antibodies in CHO-PAb-QS cells would collectively activate the endoplasmic reticulum UPR. Concurrently, the expression levels of most chaperone molecules in the ER of CHO-PAb-QS cells were markedly elevated, including BiP, heat shock protein Hsp70, and heat shock protein Hsp90, among others. The increased synthesis of these ER chaperone molecules was hypothesized to alleviate ER stress and maintain ER homeostasis. In the ubiquitin-dependent ERAD pathway, the expression of several genes in the protein degradation system involved in the ubiquitin–proteinase system was decreased, such as the OS9 gene expressing ER lectin, the p97 gene expressing valosin-containing protein, and the Nploc4 gene expressing ubiquitin recognition factor. It was hypothesized that this was due to the rapid folding of disulfide bonds in CHO-PAb-QS cells, resulting in fewer unfolded and misfolded proteins.

### 3.5. Analysis of Content of ER and Golgi Apparatus

This experiment used ER-Tracker Red probes and Golgi-Tracker Red probes to stain the cells’ ER and Golgi apparatus. The cells’ fluorescence intensity and average fluorescence intensity were analyzed by flow cytometry to determine the contents of the ER and Golgi apparatus. To ascertain whether antibody expression would increase ER and Golgi apparatus content, in addition to CHO-PAb and CHO-PAb-QS cells, CHO-K1 and CHO-QS cells were employed as controls for simultaneous experiments. Among them, the CHO-QS cell line was established based on the CHO-K1 cell line, which stably co-expressed EQK and survivin protein. The experimental results demonstrated no significant change in the volume size of the four cell lines (Figure 5A). However, according to the fluorescence intensity analysis diagram (Figure 5B,C) and the average fluorescence intensity of cells (Figure 5D,E) of flow cytometry analysis, the ER and Golgi apparatus contents of CHO-QS and CHO-PAb cells were significantly higher than those of CHO-K1 and CHO-PAb cells. The statistical results in the figure indicate that the average fluorescence intensity of CHO-PAb and CHO-PAb-QS cells increased by 7.9% and 5.3%, respectively, compared to CHO-K1 and CHO-QS cells. However, the average fluorescence intensity of CHO-QS and CHO-PAb-QS cells was 97.9% and 92.9% higher, respectively, than that of CHO-K1 and CHO-PAb cells. The Golgi apparatus changes are similar to those in the ER. The mean fluorescence intensity of CHO-PAb and CHO-PAb-QS cells was 13.4% and 1.2% higher than that of CHO-K1 and CHO-QS, respectively. The average fluorescence intensity of CHO-QS and CHO-PAb-QS cells was 69.5% and 50.5% higher than that of CHO-K1 and CHO-PAb cells, respectively.

Although the enhanced average fluorescence intensity could not fully reflect the change in ER and Golgi content, it did provide a clear indication of the increasing trend in ER and Golgi content observed in CHO-QS and CHO-PAb-QS cells. The fluorescence intensity of the ER and Golgi apparatus in cells expressing antibodies (CHO-PAb and CHO-PAb-QS) was slightly higher than that of cells not expressing antibodies (CHO-K1 and CHO-QS), indicating that the overexpression of antibodies would have a minimal effect on the ER and Golgi apparatus. Cells overexpressing the EQK protein (CHO-QS and CHO-PAb-QS) exhibited a significantly elevated ER and Golgi apparatus content. Overexpressed EQK is located in the ER, which not only enhances the folding capacity of the ER but also causes stress to the ER. To cope with the increased protein processing and the UPR, the ER increases its load capacity by increasing its capacity. Transcriptomic analysis of the ER-Golgi vesicular transport pathway (KEGG pathway map04130) revealed the upregulation of multiple transporters between the ER and Golgi (Figure 5F), such as Stx17, Bet1, Sec22, and Bos1. This result indicated that due to the ER’s enhanced folding and processing capacity in CHO-PAb-QS cells and the increased protein synthesis rate, there was an increased demand for vesicle transporters to transport mature proteins from the ER to the Golgi apparatus. Consequently, the expansion of the Golgi apparatus capacity and enhancement of protein secretion flux were achieved, ultimately leading to improved antibody production efficiency.

### 3.6. Comparison of Anti-Apoptosis Ability of Cells

In this experiment, puromycin and tunicamycin were employed to induce cell apoptosis, and the apoptosis rate was quantified by flow cytometry. After administering puromycin and tunicamycin to cells for 48 h, the CHO-PAb and CHO-PAb-QS cells exhibited varying degrees of apoptosis (Figure 6A). The apoptosis rate of CHO-PAb and CHO-PAb-QS cells under puromycin treatment was 32.6% and 17.7%, respectively, and the apoptosis rate of CHO-PAb-QS cells was 45.7% lower than that of CHO-PAb cells. Following the administration of tunicamycin, the apoptosis rate of CHO-PAb and CHO-PAb-QS cells was 24.7% and 12.9%, respectively, and the apoptosis rate of CHO-PAb-QS cells was 47.8% lower than that of CHO-PAb cells (Figure 6B). Caspase-3 is a crucial end-cut enzyme in the apoptotic process, and survivin can directly inhibit the activation of caspase-3, preventing apoptosis. Therefore, the inhibitory effect of survivin on apoptosis can be evaluated by detecting intracellular caspase-3 activity. The results demonstrated that puromycin and tunicamycin significantly increased the intracellular caspase-3 activity of the two cell lines. In the presence of puromycin, the caspase-3 activity of CHO-PAb and CHO-PAb-QS cells was 15.86 μmol/mg and 11.32 μmol/mg, respectively, and the caspase-3 activity of CHO-PAb-QS cells was 28.6% lower than that of CHO-PAb cells. Under the treatment of tunicamycin, the caspase-3 activity of CHO-PAb and CHO-PAb-QS cells was 13.31 μmol/mg and 7.88 μmol/mg, respectively, and the caspase-3 activity of CHO-PAb-QS cells was 40.8% lower than that of CHO-PAb cells (Figure 6C).

Furthermore, analysis of the KEGG pathway of the mammalian apoptosis pathway (map04215) revealed that the Fas-associated with death domain protein (FADD) and caspase-related pathways were downregulated (Figure 6D). These findings indicate that CHO-PAb-QS cell lines overexpressing survivin exhibit enhanced anti-apoptotic capabilities compared to CHO-PAb cells under the apoptotic induction of puromycin and tunicamycin.

### 3.7. Assessment of Intracellular Redox Status

Total antioxidant capacity was measured to reflect the ability of cells to fight free radicals and oxidative stress. The results showed that the total antioxidant capacity of CHO-K1, CHO-PAb, and CHO-PAb-QS cells was 0.954 nmol/mg, 0.662 nmol/mg, and 0.598 nmol/mg, respectively (Figure 7A). Compared with CHO-K1 cells, the total antioxidant capacity of CHO-PAb cells and CHO-PAb-QS cells decreased significantly to 30.6% and 37.3%, respectively, which was caused by the consumption of intracellular antioxidants during the synthesis of large amounts of antibodies by these two antibody-producing strains. The enhancement of the disulfide bond folding capacity in CHO-PAb-QS cells must consume more energy material in the cell and produce more ROS, so it is not surprising that the total antioxidant capacity of CHO-PAb-QS cells is 6.7% lower than that of CHO-PAb cells. Since the antioxidant capacity of CHO-PAb-QS was weakened and the oxidation level was increased, the redox status of CHO-PAb-QS should be further investigated.

Reduced glutathione (GSH) is involved in various redox reactions through its sulfhydryl group (-SH), which can remove free radicals and other oxidants, thereby protecting the cell from oxidative damage. Two GSH molecules are dehydrogenated by sulfhydryl groups to form an oxidized glutathione (GSSG) molecule, thereby maintaining the intracellular redox balance through reversible redox reactions between GSH and GSSG. Therefore, the redox status of CHO-PAb cells and CHO-PAb-QS cells can be investigated by measuring the cells’ total glutathione (tGSH) and depending on the ratio of GSH to GSSG. The results demonstrated that the tGSH concentrations of CHO-K1 cells, CHO-PAb cells, and CHO-PAb-QS cells were 409.2 nmol/g, 464.7 mol/g, and 485.3 nmol/g, respectively (Figure 7B). A comparison of the tGSH concentration of CHO-PAb and CHO-PAb-QS cells with that of wild-type CHO-K1 cells revealed an increase of 13.6% and 18.6%, respectively. This indicates that to clear the reactive oxygen species produced during antibody synthesis, CHO-PAb and CHO-PAb-QS cells upregulate tGSH, thus maintaining redox capacity.

The cellular redox balance is maintained primarily through the reversible redox reaction between GSH and GSSG. The reduced glutathione content (GSH/tGSH) of CHO-K1 cells, CHO-PAb cells, and CHO-PAb-QS cells was 94.3%, 88.8%, and 85.9%, and the GSH/GSSG ratio was 16.7, 7.9 and 6.1, respectively (Figure 7C). Compared to CHO-K1 cells, the GSH percentage and GSH/GSSG ratio of CHO-PAb and CHO-PAb-QS cells slightly declined. This suggests that in CHO-PAb and CHO-PAb-QS cells, the folding requirement of the antibody increased the consumption of reduced GSH, an increase in the ratio of GSSG, and a significant increase in the intracellular oxidation level, which caused an alteration in the redox state. The higher protein folding and antibody production efficiency of CHO-PAb-QS cells inevitably results in an increased load on intracellular energy metabolism, producing more reactive oxygen species. CHO-PAb-QS cells consume more reduced GSH to maintain redox homeostasis, and therefore, the GSH percentage and GSH/GSSG ratio are observed to be lower in CHO-PAb-QS cells than in CHO-PAb cells.

## 4. Discussion

In this study, we constructed the antibody cell line CHO-PAb-QS co-overexpressing HsQSOX1b and survivin based on the CHO-PAb cell line stably expressing the pembrolizumab antibody, with the starting point of improving the folding efficiency of the protein disulfide bond and the cellular anti-apoptotic ability. The results of the batch suspension culture demonstrated that the CHO-PAb-QS cell line exhibited superior characteristics, including a longer culture life, higher antibody accumulation, and enhanced unit productivity. In CHO-PAb and CHO-PAb-QS cells, antibody folding efficiency, ER stress markers, ER and Golgi apparatus contents, anti-death ability, and redox status were measured and evaluated to reveal the mechanism of HsQSOX1b and survivin co-expression affecting the production efficiency of CHO cells.

QSOX1 is a chimeric enzyme possessing disulfide bond formation and disulfide transfer capacity [45]. In vitro experiments have demonstrated that QSOX1 can catalyze the direct oxidation of various unfolded proteins at 500–2000 disulfides per minute [39]. Although QSOX1 is detected in the mammalian ER, it lacks an ER retention sequence, accumulates mainly on the Golgi apparatus, and is secreted out of the cell [46]. Hence, the contribution of QSOX1 to the folding of the ER disulfide bond remains unclear. Previous studies have demonstrated that the overexpression and localization of HsQSOX1b in the ER could significantly enhance the production of the Gaussia luciferase (GLuc) model protein [43]. Consequently, this study applied this approach to CHO antibody-producing cell lines to improve the efficiency of antibody folding. In this study, we observed that the proportion of fully folded antibodies (HC-LC)_2_ in CHO-PAb-QS cells was approximately twice that of CHO-PAb cells. When BFA was employed to impede the transfer and secretion of antibodies from the ER, the (HC-LC)_2_ in CHO-PAb-QS cells increased from 40.8% to 73.5%, representing a 32.65% increase, which was considerably higher than that observed in CHO-PAb cells. These results indicate that the overexpression of CHO-PAb-QS containing HsQSOX1b has a higher disulfide bond folding efficiency, promoting protein synthesis efficiency and secretion. In addition, the suspension batch culture experiment results demonstrated that the antibody accumulation and productivity of CHO-PAb-QS cells were increased by 52% and 45% compared to CHO-PAb cells.

The ER and the Golgi apparatus are two closely related organelles within the cell, which are involved in synthesizing, processing, and transporting secreted proteins. A significant number of studies have indicated that an increase in the content of the ER and Golgi apparatus frequently accompanies an increase in cell production. For instance, the overexpression of stearoyl CoA desaturase 1 (SCD1) and sterol regulatory element binding factor 1 (SREBF1) has been shown to regulate the lipid abundance required for ER expansion, which in turn increases cell secretion production [47]. Similarly, the overexpression of B lymphocyte-induced maturation protein-1 (BLIMP1) and X-box binding protein 1 (XBP1) has been demonstrated to induce ER and Golgi apparatus expansion, thereby improving overall productivity [48]. Therefore, we conducted a study to investigate changes in ER and Golgi content of cells. We discounted the hypothesis that increased ER content in CHO cells was responsible for the overexpression of mAbs, given that the observed rise in ER content in CHO-PAb was relatively small. The markedly enlarged ER of CHO-QS cells that did not express antibodies led us to believe that HsQSOX1b localized to the ER caused ER expansion, given that survivin had little relationship with the ER. No direct evidence of ER expansion or lipid regulation by QSOX has been reported. We are more inclined to speculate that HsQSOX1b overexpression and residence in the ER will cause an unfolded reaction, resulting in ER expansion. The increased capacity of the ER can accommodate more new proteins for processing in the ER, further improving the cell’s overall protein translation speed and productivity. More mature proteins are transported to the Golgi apparatus, which promotes the expansion of the Golgi apparatus capacity and the improvement of the transport capacity, ultimately enhancing the efficiency of antibody production.

Survivin is a small, multifunctional, and structurally unique member of the apoptosis protein inhibitor family. It is crucial in inhibiting the Bax- and Fas-induced apoptosis pathways and protects cells against caspase-dependent and non-caspase-dependent apoptosis [49,50]. In the course of cell batch culture, it was observed that the cell viability of CHO-PAb-QS cells was higher than that of CHO-PAb cells, and the cell viability period was extended by two days, showing higher anti-apoptosis. At the same time, we treated CHO-PAb cells and CHO-PAb-QS cells with puromycin and tunicamycin. The binding of puromycin to ribosomes prevents the peptide bond elongation, while tunicamycin inhibits the normal glycosylation of proteins, both resulting in ER stress. The results demonstrated that the apoptosis rate of CHO-PAb-QS cells was reduced by 45.7% and 47.8%, and the intracellular caspase-3 activity was decreased by 28.6% and 40.8%. These findings indicate that the CHO-PAb-QS cell line exhibits robust resistance to apoptosis induced by ER stress.

The total antioxidant capacity and GSH-related data of CHO-K1, CHO-PAb, and CHO-PAb-QS cells were examined. Due to the high demand for intracellular redox reactions in CHO-PAb and CHO-PAb-QS cells for antibody synthesis, the results demonstrated that the total antioxidant capacity of these two cells was significantly lower than that of CHO-K1 cells. Furthermore, it is not unexpected that CHO-PAb-QS cells exhibited a 6.7% reduction in total antioxidant capacity relative to CHO-PAb cells, given their requirement for the overexpression of functional proteins and enhanced protein folding. The ratio of GSH/GSSG can be used as an indicator to evaluate the redox status of cells. When the ratio of GSH/GSSG is elevated, the concentration of GSH in cells is relatively high, and the concentration of GSSG is relatively low, which usually indicates that cells are in a relatively reduced state and have good antioxidant capacity. On the contrary, it may indicate that the cells have been damaged by oxidative stress and require additional antioxidants to protect them from further damage. The results demonstrated that the total tGSH in CHO-PAb cells and CHO-PAb-QS cells was higher than that in CHO-K1 cells, indicating that these two antibody-producing cells have a higher redox demand, resulting in upregulated total glutathione expression. The GSH/GSSG ratio represents the redox state of the cells. The test results demonstrated that the GSH/GSSG ratio of CHO-PAb-QS cells was 6.1, representing a 22.8% reduction compared to CHO-PAb cells. In normal circumstances, GSH accounts for 90–95% of the tGSH, with less than 70% resulting in significant oxidative stress-related damage. The GSH/tGSH ratios of CHO-PAb cells and CHO-PAb-QS cells were 88.8% and 85.9%, respectively. It can be speculated that the two antibody-producing cells generated minor oxidative stress but did not cause significant changes in cell redox status. We contend that the judicious addition of supplementary antioxidants to the medium can effectively address the issue of mild oxidative stress. The addition of antioxidants to cell media to reduce oxidative stress is an accessible and effective method supported by numerous studies [51]. Lipoic acid, for instance, reduces lipid peroxidation, removes ROS, and chelates iron and copper, thereby reducing cell death [52]; pyruvate has also been shown to remove H_2_O_2_ in cell media [53]; vitamin E is known for its ROS-clearing properties and ability to resist lipid peroxidation [54]; baicalein has been shown to reduce ROS levels and inhibit the activity of transcription factors involved in ER stress responses by interacting with BiP and CHOP [55]. Other laboratory members are researching how to prevent oxidative stress in CHO-PAb-QS cells through taurine to enhance further CHO-PAb-QS cells’ productivity (results not yet published).

## 5. Conclusions

In conclusion, the co-overexpression of HsQSOX1b and survivin improved the protein synthesis in CHO-PAb-QS cells. This was achieved by enlarging the ER load, upregulating UPR activation intensity, inhibiting caspase-3 activation, and thus resisting cell apoptosis. CHO-PAb-QS cell lines demonstrated longer cell viability in batch culture and exhibited higher antibody accumulation and productivity. Consequently, the strategy of co-overexpression of HsQSOX1b and survivin can enhance the disulfide bond folding ability and anti-apoptosis ability of CHO cells, optimize the antibody production process of CHO cells, and elevate the antibody production efficiency of CHO cells, thereby establishing it as a superior production platform for biopharmaceutical manufacturing.

## Figures and Tables

**Figure 1 cells-13-01481-f001:**
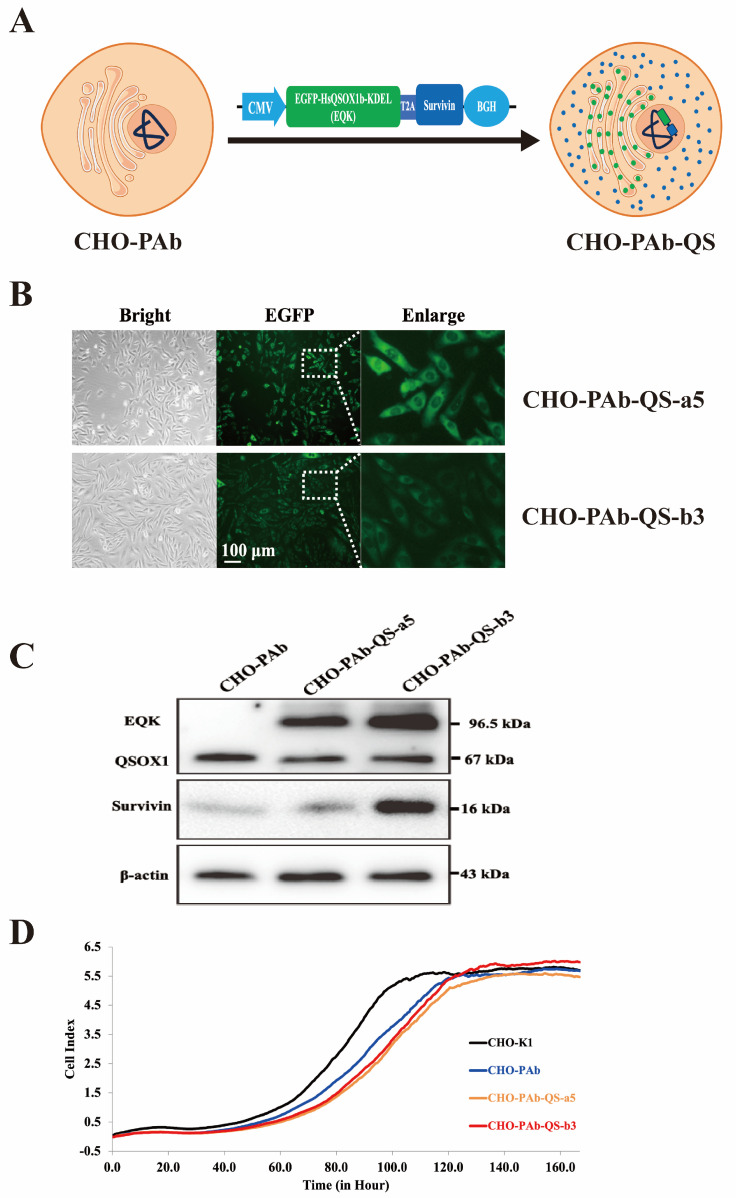
Development and characterization of EQK and survivin co-overexpressing CHO-PAb-QS cell lines. (**A**) Schematic diagram of CHO-PAb-QS cells overexpressed with EQK and survivin. (**B**) Microscope view of two CHO-PAb-QS monoclonal cell lines. CHO-PAb-QS-a5 and CHO-PAb-QS-b3 were the two positive monoclonal cells screened. In the fluorescence excitation view, HsQSOX1b protein in EQK protein fused with EGFP protein, which made the ER show green fluorescence. (**C**) Overexpression of EQK and survivin using Western blot. (**D**) The RTCA instrument measured the proliferation curves of cell lines.

**Figure 2 cells-13-01481-f002:**
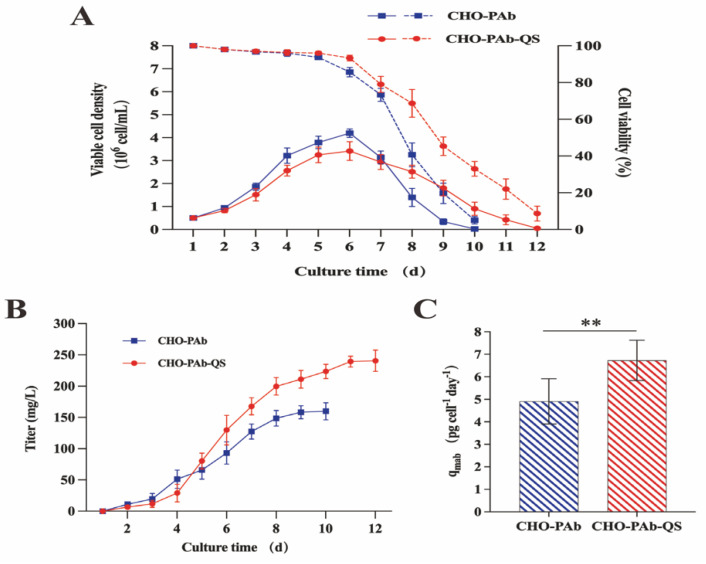
Characteristics of CHO-PAb and CHO-PAb-QS antibody production strains cultured in batches. (**A**) Viable cell density and cell viability statistics of CHO-PAb and CHO-PAb-QS cells during batch culture. The solid line represents the active cell density, and the dashed line represents the cell viability. (**B**) Antibody accumulation statistics of CHO-PAb and CHO-PAb-QS cells. (**C**) Calculation of unit productivity of CHO-PAb and CHO-PAb-QS cells. The unit productivity of CHO-PAb and CHO-PAb-QS cells was calculated based on the number of live cells and antibody accumulation on day 6 of the culture. Data were expressed as mean ± SD, n = 3, ** *p* < 0.01.

**Figure 3 cells-13-01481-f003:**
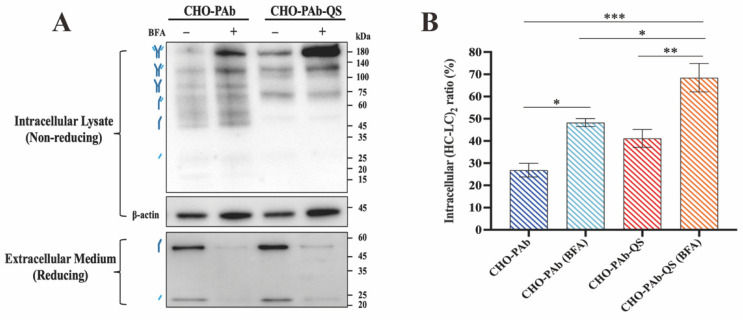
Analysis of the folding state of CHO-PAb and CHO-PAb-QS cell folding antibodies. (**A**) Western blot analysis of intracellular antibody status and extracellular secretion. Intracellular antibody assembly and folding were analyzed using non-reducing but denaturing SDS-PAGE in the presence of normal antibody secretion by cells and the presence of inhibition of antibody secretion using BFA. (**B**) Statistics on the proportion of fully folded antibodies (HC-LC)_2_ to total antibodies. Data were expressed as mean ± SD, n = 3, * *p* < 0.05, ** *p* < 0.01, *** *p* < 0.001.

**Figure 4 cells-13-01481-f004:**
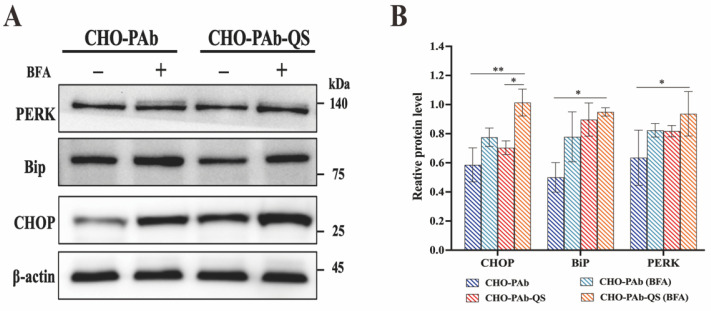
Expression of endoplasmic reticulum stress marker proteins in CHO-PAb and CHO-PAb-QS cells. (**A**) The levels of endoplasmic reticulum stress marker proteins PERK, BiP, and CHOP were detected by Western Blot. (**B**) Relative expression analysis of endoplasmic reticulum stress marker proteins. Data were expressed as mean ± SD, n = 3, * *p* < 0.05, ** *p* < 0.01.

**Figure 5 cells-13-01481-f005:**
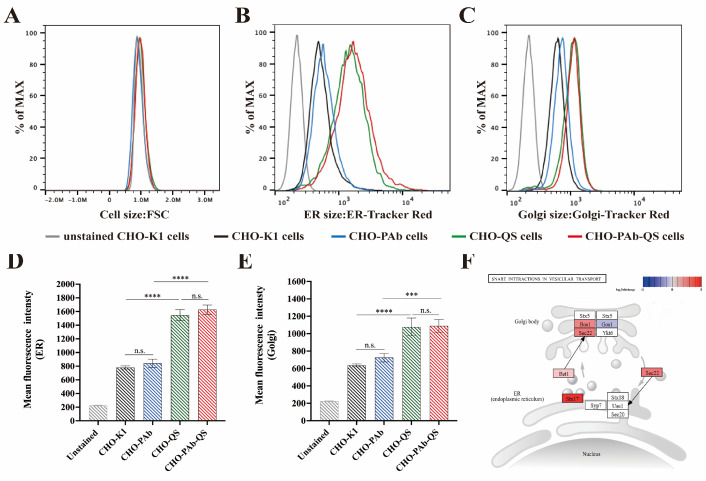
Analysis of the ER and Golgi apparatus content by flow cytometry. (**A**) Forward scattered light FSC was used to measure cell size. (**B**) The fluorescence intensity of cells labeled with ER-tracker Red was detected. (**C**) The average fluorescence intensity of the cells was calculated according to the flow cytometry results in Figure (**B**). (**D**) The fluorescence intensity of cells labeled with Golgi-tracker Red was detected. (**E**) The average fluorescence intensity of cells according to the flow cytometry analysis results in Figure (**D**). Gray represents unstained CHO-K1 cells, black represents stained CHO-K1 cells, blue represents stained CHO-PAb cells, green represents stained CHO-QS cells, and red represents stained CHO-PAb-QS cells. The genes are color-coded according to their log_2_Foldchange values (*p*-value < 0.05), where white-marked genes are *p*-value ≥ 0.05 or genes that have not been sequenced. (**F**) KEGG pathway of CHO-PAb-QS ER-Golgi vesicle transport pathway (map04130). The genes are color-coded according to their log_2_Foldchange values (*p*-value < 0.05), where white-marked genes are *p*-value ≥ 0.05 or genes that have not been sequenced. Data were expressed as mean ± SD, n = 3, *** *p* < 0.001, **** *p* < 0.0001, n.s. means not significant.

**Figure 6 cells-13-01481-f006:**
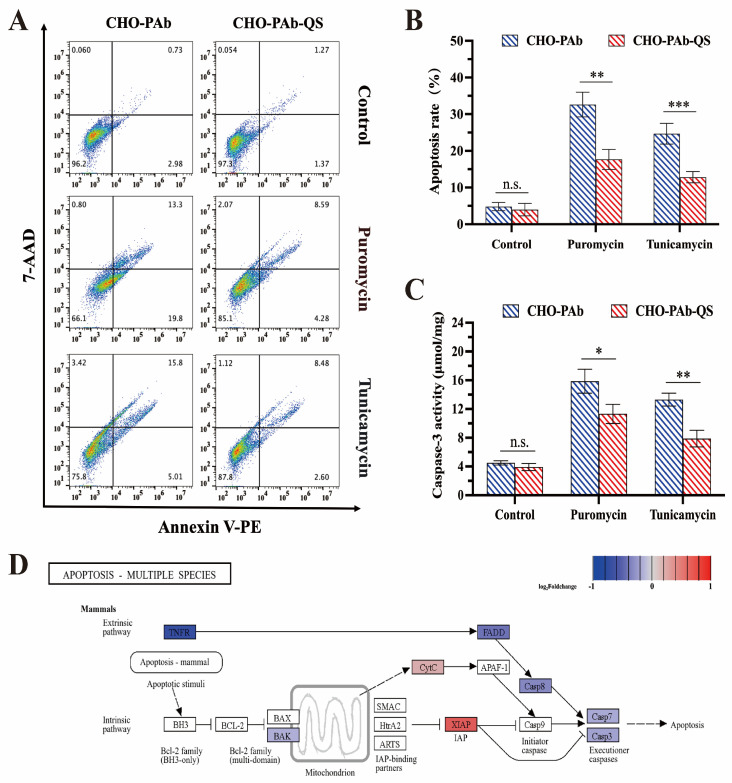
Analysis of apoptosis of CHO-PAb and CHO-PAb-QS cells. (**A**) Flow cytometry detected the apoptosis rate of CHO-PAb and CHO-PAb-QS cells after treatment with puromycin and tunicamycin for 48 h. (**B**) Statistical results of apoptosis rate. (**C**) Caspase-3 activity of CHO-PAb and CHO-PAb-QS cells treated with puromycin and tunicamycin for 48 h. (**D**) KEGG pathway of CHO-PAb-QS cells apoptosis pathway (map04215). The genes are color-coded according to their log_2_Foldchange values (*p*-value < 0.05), where white-marked genes are *p*-value ≥ 0.05 or genes that have not been sequenced. Data were expressed as mean ± SD, n = 3, * *p* < 0.05, ** *p* < 0.01, *** *p* < 0.001, n.s. means not significant.

**Figure 7 cells-13-01481-f007:**
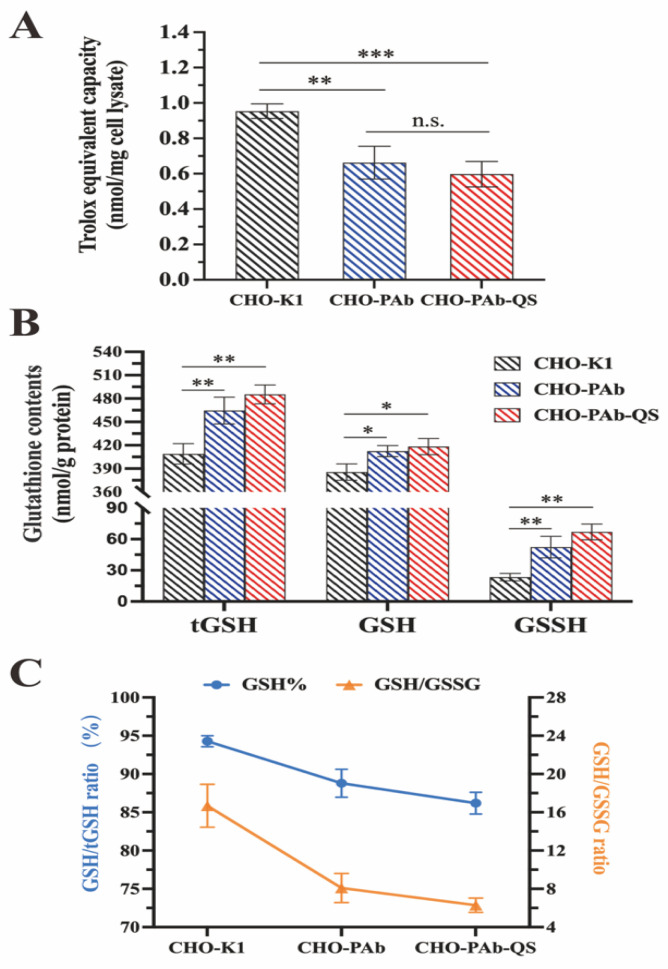
Detection of oxygen reduction status in cells. (**A**) Detection of total antioxidant capacity of CHO-K1, CHO-PAb, and CHO-PAb-QS cells. (**B**) The content detection of tGSH, GSH, and GSSG. The tGSH stands for total glutathione within cells, including GSH and GSSG. (**C**) Statistics of GSH/tGSH and GSH/GSSG ratios. * *p* < 0.05, ** *p* < 0.01, *** *p* < 0.001, n.s. means not significant.

## Data Availability

The original contributions presented in the study are included in the article/Appendix A, further inquiries can be directed to the corresponding author.

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
