# Peer review of "Enhancing the Antibody Production Efficiency of Chinese Hamster Ovary Cells through Improvement of Disulfide Bond Folding Ability and Apoptosis Resistance"

_cells, 2024, doi:10.3390/cells13171481_

Round 1

Reviewer 1 Report

Comments and Suggestions for Authors

The article "Enhancing the antibody production efficiency of CHO cells through improvement of disulfide bond folding ability and apoptosis resistance" by Zhang et al. (2024) explores the enhancement of antibody production efficiency in Chinese hamster ovary (CHO) cells by co-overexpressing HsQSOX1b and Survivin proteins. The study demonstrates that this genetic modification improves disulfide bond folding and apoptosis resistance, leading to increased antibody yield and productivity. The CHO-PAb-QS cell line, developed by the researchers, showed a 52% increase in antibody accumulation and a 45% increase in productivity compared to the parent CHO-PAb cell line. The strengths of the research include the innovative approach of using HsQSOX1b and Survivin co-overexpression, the comprehensive methods employed, such as plasmid construction, cell culture, Western blotting, flow cytometry, and transcriptome analysis, and the detailed analysis of cell viability, antibody production, ER stress marker expression, and redox status. However, the study's limited scope and potential bias are weaknesses that future research could address by exploring  different antibodies. The results are well-presented, showing increased productivity and enhanced folding efficiency, with the authors accurately interpreting the data to conclude that co-overexpression of HsQSOX1b and Survivin enhances antibody production efficiency in CHO cells. This study makes a significant contribution to biotechnology and cellular engineering by providing a viable strategy to enhance antibody production, which could be applied to other antibodies. For future research, broader applications, mechanistic studies, and investigations into the long-term stability and productivity of the modified cell lines in industrial bioreactors are recommended. Overall, the article is a well-executed study that addresses key challenges in antibody production using CHO cells, and despite some limitations, its contributions are significant, paving the way for future research and industrial applications.

Author Response

Comments: The article "Enhancing the antibody production efficiency of CHO cells through improvement of disulfide bond folding ability and apoptosis resistance" by Zhang et al. (2024) explores the enhancement of antibody production efficiency in Chinese hamster ovary (CHO) cells by co-overexpressing HsQSOX1b and Survivin proteins. The study demonstrates that this genetic modification improves disulfide bond folding and apoptosis resistance, leading to increased antibody yield and productivity. The CHO-PAb-QS cell line, developed by the researchers, showed a 52% increase in antibody accumulation and a 45% increase in productivity compared to the parent CHO-PAb cell line. The strengths of the research include the innovative approach of using HsQSOX1b and Survivin co-overexpression, the comprehensive methods employed, such as plasmid construction, cell culture, Western blotting, flow cytometry, and transcriptome analysis, and the detailed analysis of cell viability, antibody production, ER stress marker expression, and redox status. However, the study's limited scope and potential bias are weaknesses that future research could address by exploring  different antibodies. The results are well-presented, showing increased productivity and enhanced folding efficiency, with the authors accurately interpreting the data to conclude that co-overexpression of HsQSOX1b and Survivin enhances antibody production efficiency in CHO cells. This study makes a significant contribution to biotechnology and cellular engineering by providing a viable strategy to enhance antibody production, which could be applied to other antibodies. For future research, broader applications, mechanistic studies, and investigations into the long-term stability and productivity of the modified cell lines in industrial bioreactors are recommended. Overall, the article is a well-executed study that addresses key challenges in antibody production using CHO cells, and despite some limitations, its contributions are significant, paving the way for future research and industrial applications.

Response: We thank the reviewer for finding interest in our manuscript and for the accurate summary of the paper's results.

Reviewer 2 Report

Comments and Suggestions for Authors

Todays standard of yield from CHO cells are in the 5 - 15 g/L range WITHOUT ANY CELL ENGINEERING.  Thus a paper that improves yield from several hundred milligrams to something slightly higher is not relevant today. Both folding issues with antibodies and apoptosis intervention have been shown to be of limited value, if at all. This reviewer does not question the overall quality of research done, however the authors have obviously too many years outside of the field and are REINVENTING problems which are not existing anymore...

Overall the paper is well written. It is probably a fair conclusion that the authors are latecomers to the field and had not the chance to follow the overall progress of the field... Apoptosis and other issues of folding can be addressed today with better media and feed (all chemically defined today) and with controlling the processes in bioreactors in a smart way. 

This reviewer regrets therefor to refuse this paper for acceptance. 

Comments on the Quality of English Language

no further comment necessary... 

Author Response

Comments: Todays standard of yield from CHO cells are in the 5 - 15 g/L range WITHOUT ANY CELL ENGINEERING.  Thus a paper that improves yield from several hundred milligrams to something slightly higher is not relevant today. Both folding issues with antibodies and apoptosis intervention have been shown to be of limited value, if at all. This reviewer does not question the overall quality of research done, however the authors have obviously too many years outside of the field and are REINVENTING problems which are not existing anymore...

Overall the paper is well written. It is probably a fair conclusion that the authors are latecomers to the field and had not the chance to follow the overall progress of the field... Apoptosis and other issues of folding can be addressed today with better media and feed (all chemically defined today) and with controlling the processes in bioreactors in a smart way. 

This reviewer regrets therefor to refuse this paper for acceptance. 

Response:  Thank you for your feedback and for raising your concern regarding the necessity of our study. We would like to address your comment and explain why our study still contributes to the existing knowledge and advances the CHO cells field.

Cell engineering is potentially important for CHO cells to break through the yield bottleneck. We have to admit that in the current field of biological drug production, the use of productive CHO cells has not been transformed by cell engineering, and cell culture technology has been very mature, so the drug yield of cells has also greatly improved. However, there is a limit to the external culture conditions to improve the production performance of CHO cells. Although cell engineering has not made a great breakthrough in the industrial production of CHO cells, we cannot deny that it is an important means to transform CHO to achieve the potential of cells. There are many domestic and foreign research groups in cell engineering to study ways to improve cell production efficiency, such as protein transcription and translation, post-translational modification, and protein secretion.

Improving disulfide bond folding and anti-apoptotic ability is a potential strategy to increase the production of complex recombinant protein drugs. Although cell drug production has also been greatly improved, biological drugs are expensive. In addition to research and development costs being a huge input, the complexity and low yield of the production process are also important. Especially with the continuous breakthrough of biotechnology and medical research, more and more biological drugs have been developed and produced. Most of these biological drugs are recombinant proteins and antibodies with complex structures, and complex folding processes are required in the production process. Our research group has 20 years of experience in cancer drug research and development and found that QSOX and Survivin genes are highly expressed in cancer cells. Therefore, we innovatively put these two genes into CHO cells for exploration, hoping to improve the protein folding ability and anti-apoptosis ability of CHO cells and contribute a strategy for increasing the yield of CHO antibodies.

Our laboratory is biased towards basic research, and cutting-edge information on the industrial production of CHO cells is mainly obtained from published articles, magazines and conferences. Production enterprises have their own research and development team, usually the specific situation of production and confidential research technology, so there are barriers to university research laboratories. We are impressed by the reviewers' knowledge of the CHO cell industrial application production industry. If possible, we sincerely invite you to have academic exchanges with my research group so that we can better grasp the key issues in the CHO production process and conduct targeted research.

We hope this clarification addresses your concerns regarding the necessity of our study.  Again, we appreciate your time and comments.

Reviewer 3 Report

Comments and Suggestions for Authors

“Enhancing the antibody production efficiency of CHO cells through improvement of disulfide bond folding ability and apoptosis resistance” by Chen Zhang, Yunhui Fu, Wenyun Zheng, Feng Chang, Yue Shen, Jinping Niu, Yangmin Wang, Xingyuan Ma (Manuscript ID: cells-3111015).

This manuscript investigated the effects of overexpressing HsQSOX1b and Survivin proteins in a stable CHO cell line on enhancing antibody production, building on the findings from its previous publication (Wang, W. et al, Ma, X. ACS Synthetic Biology 2018). The authors confirmed the anti-apoptosis benefit induced by ER stress as well as the milder oxidative status in the resulting cell line. Consequently improvement of antibody expression titer was observed. The data also uncovered the differences in the folding state of antibody HC-LC complexes, suggesting a better efficiency in disulfide bond folding with the overexpression of the two cofactors.  The study further investigated the impacts on ER stress markers, ER/Golgi apparatus contents, and redox status. The topic of the manuscript is therapeutically interesting and the findings are original. Prior to the acceptance, there are a few minor points needed to be addressed:

1)     Several figures (Figure 1A&B; Figure 5F, Figure 6D) are illegible and their quality needs to be improved.

2)     In Figure 1A, is there a promoter between EQK and Survivin genes? Figure 2B is too tiny, and need an ER-tracker stain.

3)     For Figure 5A-C, what are those lines representing, CHO-K1, Pab, QS, or Pab-QS?

4)     Page 9, line 388, in what day the samples were collected for Western Blot analysis?

5)     Page 17, line 690-693, the reported ratios of GSH/tGSH (85%-95%) in the study were high. Does it mean the redox status is largely being reduced?

Author Response

Thank you for your comments concerning our manuscript. Those comments are valuable and helpful for revising and improving our paper and have important guiding significance to our research. We have studied the comments carefully and have made corrections, which we hope meet with your approval. The revised portions are marked in yellow in the manuscript.

Comments 1: Several figures (Figure 1A&B; Figure 5F, Figure 6D) are illegible and their quality needs to be improved.

Response 1: Thank you for your feedback and suggestions. We have carefully reviewed the original figures and made the necessary revisions to improve their clarity. For Figure 1A&B, we modified and rearranged them and enlarged the images to show the experimental results more clearly. For Figure 5F and Figure 6D, these two data were analyzed and downloaded from the KEGG Pathway Database website, and there is no way for us to change the size of the data displayed in the figures. However, we enlarged the images as much as possible and increased the output resolution to 600 dpi to make them more transparent.

Comments 2: In Figure 1A, is there a promoter between EQK and Survivin genes? Figure 2B is too tiny, and need an ER-tracker stain.

Response 2: Thank you for your comments. 1)In Figure 1A, there was no promoter between EQK and Survivin genes. We linked EQK and Survivin genes with the T2A polypeptide gene sequence. T2A is an 18-amino acid polypeptide encoded by 54 bases, with the last three amino acids being Pro-Gly-Pro. Because eukaryotic ribosomes cannot form peptide bonds between the two amino acid residues Gly and Pro, the translated peptide chain will automatically break to form two separate polypeptide chains. We applied this T2A to the construction of EQK-T2A-Survivin to obtain two separate proteins that perform their respective intracellular functions without needing a separate translation frame. 2)Although the comment refers to Figure 2B, we guess it should refer to Figure 1B. If we are wrong, please give us feedback again. We have enlarged the partial image of Figure 1B to make the endoplasmic reticulum markers clearer.

Comments 3: For Figure 5A-C, what are those lines representing, CHO-K1, Pab, QS, or Pab-QS?

Response 3: Thank you for your comments. In Figure 5A-C, the grey line represents unstained CHO-K1 cells, the black line represents stained CHO-K1 cells, the blue line represents stained CHO-PAb cells, the green line represents stained CHO-QS cells, and the red line represents stained CHO-PAb-QS cells. We annotated them in Figure 5 to make it clearer.

Comments 4: Page 9, line 388, in what day the samples were collected for Western Blot analysis?

Response 4: Thank you for your comments. The sample collection time for this Western Blot analysis was 32 h after cell passage culture. We added the specific methods to Materials and Methods 2.5 in the manuscript.

Comments 5: Page 17, line 690-693, the reported ratios of GSH/tGSH (85%-95%) in the study were high. Does it mean the redox status is largely being reduced?

Response 5: Thank you for your comments. GSH is a key antioxidant in animal cells and protects cells from damage caused by oxidative stress. ‌The GSH/tGSH ratio can directly reflect the ability of cells to fight oxidative stress and cell health. Our experimental results showed that the GSH/tGSH ratios of CHO-PAb cells (88.8%) and CHO-PAb-QS cells (85.9%) were slightly lower than 90%. In normal circumstances, GSH accounts for 90-95% of the tGSH, with less than 70% resulting in significant oxidative stress-related damage. Therefore, we believe that the redox state of CHO-PAb and CHO-PAb-QS cells was reduced mainly to the reduction state.

Reviewer 4 Report

Comments and Suggestions for Authors

"When the thiol-containing substances such as tri(2-carboxyethyl) phosphine and dithiothreitol ..." Tri(2-carboxyethyl) phosphine does not contain any thiols. Perhaps "disulfide-reducing" was intended.

"with approximately 5×105 cells added" The second digit 5 should be superscripted.

"... flow cytometry was 488nm." -> 488 nm.

Line 248, 251 "absorbance of A405nm," -> absorbance at 405 nm, (?)

In Fig. 1C the bands are cut out of the gel. At some point (supplement?) the whole gel should be shown, including the size calibration ladder.

"Elisa quantified ..." -> ELISA

Materials: The expressed antibody Pembrolizumab should be described in more detail. The sole reference to [44] is inconvenient. Also, how this cell line was constructed would be interesting. Does this antibody have special properties that could be relevant in this context? Is this cell line freely available or is it subject to patent restrictions?

The abbreviation DIMA is not explained.

"enzyme-labeled goat anti-human IgG (h + L)" -> HRP-labeled goat anti-human IgG (h + l).

"The absorbance at 405nm was then detected by the microplate reader (Thermo)." -> 405 nm; please also give the exact type of the reader.

Line 634 and others: "40.82% to 73.47%" The number of digits is too high in relation to the precision of the measurements. 40.8% to 73.5% or even 41% to 73% would be more adequate.

The "Discussion" and "Conclusion" sections appear to be too long in relation to the results of the work. Therefore, some parts of the text seem to repeat themselves. The manuscript could benefit from a significant condensation of the text.

Author Response

Thank you for your comments concerning our manuscript. Those comments are valuable and helpful for revising and improving our paper and have important guiding significance to our research. We have studied the comments carefully and have made corrections, which we hope meet with your approval. The revised portions are marked in yellow in the manuscript.

Comments 1: "When the thiol-containing substances such as tri(2-carboxyethyl) phosphine and dithiothreitol ..." Tri(2-carboxyethyl) phosphine does not contain any thiols. Perhaps "disulfide-reducing" was intended.

Response 1: Thank you for your careful checking. We are sorry for our carelessness. The tri(2-carboxyethyl) phosphine (TCEP) contains no thiols. Our lab previously reported that sulfhydryl oxidases, including QSOXs, can also efficiently oxidise non-thiol reductant substrates, such as TCEP, a specific reductant for disulfide. We have revised it in the manuscript.

Comments 2: "with approximately 5×105 cells added" The second digit 5 should be superscripted.

Response 2: Thank you for your careful checking. We did not find this error in the manuscript proofreading, and we have changed it to "5×105 " in the manuscript.

Comments 3: "... flow cytometry was 488nm." -> 488 nm.

Response 3: We are sorry for our careless mistakes. Thank you for reminding us. We have changed it to "488 nm " in the manuscript.

Comments 4: Line 248, 251 "absorbance of A405nm," -> absorbance at 405 nm, (?)

Response 4: Thank you for your careful checking. We have changed it to "405 nm " in the manuscript.

Comments 5: In Fig. 1C the bands are cut out of the gel. At some point (supplement?) the whole gel should be shown, including the size calibration ladder.

Response 5: Thank you for your comments. All original images and data of western blot results have been submitted to the editorial department as required.

Comments 6: "Elisa quantified ..." -> ELISA

Response 6: Thank you for your careful checking. We changed "Elisa" to "ELISA" in the manuscript.

Comments 7: Materials: The expressed antibody Pembrolizumab should be described in more detail. The sole reference to [44] is inconvenient. Also, how this cell line was constructed would be interesting. Does this antibody have special properties that could be relevant in this context? Is this cell line freely available or is it subject to patent restrictions?

Response 7: Thank you for your comments. 1) We have added a description of CHO-PAb cells in Materials and Methods 2.2. Reference [44] reports another interesting research result from our lab: a CHO-CDbox cell platform constructed based on site-specific recombination systems. CHO-PAb cells were developed from CHO-CDbox cells by site-specific recombinase exchange. The Pembrolizumab gene was integrated as a single copy into the H11 locus (a genomic locus that does not code for any genes), giving it the property of stably expressing the Pembrolizumab. 2) We do not think that it has a special connection with folding and anti-apoptosis in this study. 3) Besides, from the point of view of scientific research, CHO-PAb cells are not patent-restricted.

Comments 8: The abbreviation DIMA is not explained.

Response 8: Thank you for your careful checking. "DIMA" refers to DIMA Biotechnology LLC. (Wuhan, China, www.dimabio.com). Our Lab purchased PD-1 antigen and pembrolizumab antibody from this company. We have modified "DIMA" to "DIMA BIOTECH" to make it more explicit in the manuscript.

Comments 9: "enzyme-labeled goat anti-human IgG (h + L)" -> HRP-labeled goat anti-human IgG (h + l).

Response 9: Thank you for your careful checking. Your suggestion is correct, "enzyme-labeled goat anti-human IgG (h + L) " should be corrected to "HRP-labeled goat anti-human IgG (h + L) ". In order to be consistent with the writing format, I finally changed it to " HRP-Goat Anti-Human IgG (H+L)" in the manuscript.

Comments 10: "The absorbance at 405nm was then detected by the microplate reader (Thermo)." -> 405 nm; please also give the exact type of the reader.

Response 10: Thank you for your careful checking. 1) We changed "405nm" to "405 nm" in the manuscript. 2) The reader, full name Multiskan SkyHigh Microplate Spectrophotometer (Catalog number: A51119700DPC), comes from Thermo Fisher Scientific Inc. We revised it in the manuscript to express it more clearly.

Comments 11: Line 634 and others: "40.82% to 73.47%" The number of digits is too high in relation to the precision of the measurements. 40.8% to 73.5% or even 41% to 73% would be more adequate.

Response 11: Thank you for your suggestion. We changed it to "40.8% to 73.5%" in the manuscript.

Comments 12: The "Discussion" and "Conclusion" sections appear to be too long in relation to the results of the work. Therefore, some parts of the text seem to repeat themselves. The manuscript could benefit from a significant condensation of the text.

Response 12: Thank you for your comment and suggestion. After thoroughly reviewing the whole text, we have reduced and compressed the discussion of repeated narratives.

Round 2

Reviewer 2 Report

Comments and Suggestions for Authors

Dear authors of the paper here reviewed by myself. 

I regret that I cannot approve the publishing of this paper, indicated already by my first review. While it is regrettable that some of the progress in our industry, leading to 10g/L and higher yields WITHOUT cell engineering has not been available or accessible to an academic lab in China, it does not provide a reason to publish a paper which reproduces publications of the past, some of them going back more than 20 years. I have personally reviewed papers on blocking apoptosis in CHO cells, decades ago, essentially with the same results as you report. Also, folding issues in CHO cells have been addressed in prior publications, but again, if vectors and codons, and protein structures are optimised, and ... most importantly, if media and feeds are improved allowing higher cell masses, then yields as quoted are possible. If you wish to publish your work, you should use a title that does not deceive the potential readers - which are mostly in the industrial environment: you may try to restructure your work so that it does not claim higher yields but just studies SPECIFIC problems, i.e. refers to your protein of interest and and shows why for this protein certain cell death inducing issues occur and why folding issues can be addressed. 

AGAIN: I am not qualifying your work, but your title addresses an audience in the biosimilar field, which looks for MUCH higher titres than what you resport (and they have been achieved, even from groups not working within the companies of the originators). 

Also, if you quote papers on reducing apoptosis in the CHO you need to look at conference proceedings as well that go back into the 1980s and 1990 (the proceedings of the European Society of Animal Cell Technolog - ESACT - are an excellent source). You will find a large amount of information there. 

I hope you understand. 

Comments on the Quality of English Language

English is acceptable 

Author Response

Comment: Dear authors of the paper here reviewed by myself. 

I regret that I cannot approve the publishing of this paper, indicated already by my first review. While it is regrettable that some of the progress in our industry, leading to 10g/L and higher yields WITHOUT cell engineering has not been available or accessible to an academic lab in China, it does not provide a reason to publish a paper which reproduces publications of the past, some of them going back more than 20 years. I have personally reviewed papers on blocking apoptosis in CHO cells, decades ago, essentially with the same results as you report. Also, folding issues in CHO cells have been addressed in prior publications, but again, if vectors and codons, and protein structures are optimised, and ... most importantly, if media and feeds are improved allowing higher cell masses, then yields as quoted are possible. If you wish to publish your work, you should use a title that does not deceive the potential readers - which are mostly in the industrial environment: you may try to restructure your work so that it does not claim higher yields but just studies SPECIFIC problems, i.e. refers to your protein of interest and and shows why for this protein certain cell death inducing issues occur and why folding issues can be addressed. 

AGAIN: I am not qualifying your work, but your title addresses an audience in the biosimilar field, which looks for MUCH higher titres than what you resport (and they have been achieved, even from groups not working within the companies of the originators). 

Also, if you quote papers on reducing apoptosis in the CHO you need to look at conference proceedings as well that go back into the 1980s and 1990 (the proceedings of the European Society of Animal Cell Technolog - ESACT - are an excellent source). You will find a large amount of information there. 

I hope you understand. 

Response: While we appreciate the reviewer’ s feedback, we respectfully disagree. We would like to explain some of our ideas and hopefully give you a better view of our article.

  1. First of all, about CHO yield.

Our laboratory is a laboratory in a university, and there is no professional cell bioreactor. Therefore, we used the traditional cell culture flask to carry out batch culture experiments, and the experimental results were mainly compared according to the yield under the same conditions and did not compare with the production from the cell bioreactor. Depending on the differences in cell lines and culture conditions, the literature reports CHO cell yields of approximately 0.1-1 mg/mL in batch culture and 1-10 mg/mL in batch replenishment culture. The yield (241 μg/mL and 159 μg/mL) in our experiment belongs to the range of batch culture 0.1-1 mg/mL. The reviewer repeatedly compared the output of fed-batch cultures with the output results of this experiment to conclude that our study is not meaningful, and we believe that such comparisons and judgments are unfair to us.

Moreover, we do not consider our title deceptive. On the contrary, we think the title of this article is appropriate, not only to provide a good overview of the study, but also to highlight the purpose and significance of the study. It was clearly stated in the paper that the study was done in the laboratory under batch culture conditions, and we focused on the increase of yield, rather than the amount of yield. We believe that readers who read this article carefully will not be misled.

  1. Secondly, the problem of the folding issues and apoptosis mentioned by reviewer.

The protein folding pathway in mammalian cells is a complex multi-step process, which mainly relies on protein disulfide isomerase (PDI) to catalyze the formation of disulfide bonds in proteins, and endoplasmic reticulum oxidoreductase 1 (Ero1) participates in the reoxidation of PDI to make it play a continuous role. Combining the previous literature on PDI and EroI, it can be found that CHO cell production may depend on the EroI/PDI ratio, which is difficult to control accurately. This led to subsequent studies on the improvement of folding in CHO cells, mainly through the regulation of endoplasmic reticulum chaperone molecules and UPR-related factors indirectly promoting the folding process. QSOX1 is a sulfhydryl oxidase that possesses both disulfide bond formation and disulfide transfer capabilities, and studies have shown that QSOX1 may have a higher rate of disulfide bond generation than the PDI-EroI system. Therefore, our group has previously investigated its disulfide bond catalytic folding properties and then introduced the human quiescin sulphhydryl oxidase 1 isoform b (hsQSOX1b) into CHO cells to explore new strategies to promote folding efficiency. The future biopharmaceutical industry will be flooded with more and more complex therapeutic proteins, such as monoclonal antibodies (mAb), fusion proteins, hormones, enzymes, and coagulation factors. Most of these biopharmaceuticals are recombinant proteins and antibodies with complex structures, and complex folding processes are required in the production process. We believe that the improvement of folding function is of great importance for CHO production cells.

Indeed, as reviewer said, improving the anti-apoptosis ability of CHO cells has been paid attention by researchers in the CHO field for decades, and the inhibition of apoptosis of CHO has been explored in various ways, and good results have been obtained, including the addition of anti-apoptotic agents to the medium. Coincidentally, another important focus of our research group is therapeutic cancer research around the Survivn protein. Years of research on Survivn convinced us that it is a functional protein that is very effective in resisting apoptosis. Therefore, we selected Survivn as a functional protein that inhibits apoptosis and maintains cell homeostasis after hsQSOX1b improves endoplasmic reticulum folding performance. According to reviewer, apoptosis can already be improved by improvements in culture medium and bioreactor, and our strategy of maintaining homeostasis with Survivn is meaningless. We feel that reviewer's research thinking is not inclusive enough, and at the same time, it is unfair to our study.

  1. Third, on the issue of cell engineering to increase CHO cell production

Reviewer repeatedly emphasized that without the intervention of cell engineering, higher cell production can be achieved by culture-medium and bioreactor alone. But we don't think so. It is impossible to completely break through the limitations of CHO cells only by changes in external conditions. To meet the growing demands of the biopharmaceutical market, researchers expect to achieve higher batch productivity in a shorter time, as well as stable product quality and lower production costs. To achieve this, researchers have engineered cells that overexpress beneficial genes or suppress unfavorable genes to improve the performance of CHO manufacturing cell lines. These cell engineering approaches have classically focused on cell growth, metabolism, apoptosis, protein glycosylation, secretion, and more. Although the current research on the transformation of CHO cells has not been applied to the industrial production of mature technical results, we believe that the future breakthrough in the production of CHO cells must be inseparable from cell engineering.

Finally, we would like to state that our lab has been studying antibody expression in CHO cells for ten years. In 2017, our lab applied for a project titled ‘Using CRISPR/Cas9 double gene to edit and enhance CHO antibody expression system’ (No. 31670944) supported by the National Natural Science Foundation. We are not outside the field and reviewer seems to have misunderstood us.

We appreciate your time and consideration, and we remain open to any further suggestions or feedback you may have.